SC-JNMF: single-cell clustering integrating multiple quantification methods based on joint non-negative matrix factorization

Shiga Mikio
Seno Shigeto senoo@ist.osaka-u.ac.jp
Onizuka Makoto
Matsuda Hideo
Graduate School of Information Science and Technology, Osaka University , Osaka , Japan
Nakai Kenta
Electronic publication date: 2021 Aug 27
Publication date: 2021
Volume: 9
Electronic Location ID: e12087
Received 2021 Mar 10; Accepted 2021 Aug 7
Copyright: ©2021 Shiga et al.
Copyright year: 2021
Copyright holder: Shiga et al.
License: This is an open access article distributed under the terms of the Creative Commons Attribution License, which permits unrestricted use, distribution, reproduction and adaptation in any medium and for any purpose provided that it is properly attributed. For attribution, the original author(s), title, publication source (PeerJ) and either DOI or URL of the article must be cited.
License URL: https://creativecommons.org/licenses/by/4.0/

Keywords: Single-cell, RNA-seq, Non-negative matrix factorization, Clustering

Funding: JSPS KAKENHI, Japan 19H04207 This work was supported by JSPS KAKENHI Grant Number 19H04207, Japan. The funders had no role in study design, data collection and analysis, decision to publish, or preparation of the manuscript.

==============================
Single-cell RNA-sequencing is a rapidly evolving technology that enables us to understand biological processes at unprecedented resolution. Single-cell expression analysis requires a complex data processing pipeline, and the pipeline is divided into two main parts: The quantification part, which converts the sequence information into gene-cell matrix data; the analysis part, which analyzes the matrix data using statistics and/or machine learning techniques. In the analysis part, unsupervised cell clustering plays an important role in identifying cell types and discovering cell diversity and subpopulations. Identified cell clusters are also used for subsequent analysis, such as finding differentially expressed genes and inferring cell trajectories. However, single-cell clustering using gene expression profiles shows different results depending on the quantification methods. Clustering results are greatly affected by the quantification method used in the upstream process. In other words, even if the original RNA-sequence data is the same, gene expression profiles processed by different quantification methods will produce different clusters. In this article, we propose a robust and highly accurate clustering method based on joint non-negative matrix factorization (joint-NMF) by utilizing the information from multiple gene expression profiles quantified using different methods from the same RNA-sequence data. Our joint-NMF can extract common factors among multiple gene expression profiles by applying each NMF under the constraint that one of the factorized matrices is shared among multiple NMFs. The joint-NMF determines more robust and accurate cell clustering results by leveraging multiple quantification methods compared to conventional clustering methods, which use only a single gene expression profile. Additionally, we showed the usefulness of discovering marker genes with the extracted features using our method.

Introduction

Advances in technology have made it possible to isolate individual cells from a population of cells and to sequence their transcriptomes at the single-cell level, known as single-cell RNA-sequencing (scRNA-seq). This technology has reached a surprising level of resolution that reveals the regulation of gene expression within cells. scRNA-seq measures gene expression on a cell-by-cell basis and allows the analysis of the functions and properties of cells using this information. Many experimental protocols and computational analyses exist for scRNA-seq, and they have different goals such as differential expression analysis, cell clustering, cell classification, and trajectory reconstruction. Therefore, single-cell analysis forms a pipeline (a series of procedures) that mainly consists of two parts, quantification of gene expression and downstream analysis depends on the goals. In the quantification part of a pipeline, gene expression levels measured for each cell (raw sequence data) are converted to a matrix called the “gene expression profile”. The quantification part also forms a pipeline including read quality control, adaptor trimming, demultiplexing, deduplicating using barcodes, mapping to genome, counting transcripts.

One of the objectives of single-cell analysis is to identify cell types by applying unsupervised clustering and extract a group of characteristic genes for a specific cell type as marker genes. Clustering is a useful method for the classification and identification of unknown cell groups and discovering the diversity and subpopulation of known cell types, and it is a fundamental step in scRNA-seq data analysis. It is the key to understanding cell function and constitutes the basis of other advanced analyses. Quantification is a critical factor for the subsequent clustering of analysis results. Different quantification methods result in different gene expression values even when the same RNA-seq library is processed. The greatest difference in the quantification methods is using an alignment-based method or an alignment-free method. As examples of alignment-based methods, short reads of the RNA-seq library are aligned to a reference genome using alignment tools, such as Bowtie2 (Langmead & Salzberg, 2012) and STAR (Dobin et al., 2013). Then, the gene expression values are obtained from the results using gene expression estimation tools such as RSEM (Li et al., 2010) and Cufflinks (Trapnell et al., 2010). In contrast, alignment-free tools such as Salmon(Patro et al., 2017) and kallisto(Bray et al., 2016) estimates mRNA abundances with k-mer counting approach (pseudo-alignment for transcriptome indices). Depends on the methods and the property of the RNA-seq library, several units of gene expression values are used such as RPKM (Reads per Kilobase Million), FPKM (Fragments per Kilobase Million), TPM (Transcripts per Million), UMI (Unique Molecular Identifiers) counts, and raw read counts. In addition, there are specialized quantification pipelines for the platform, such as Cell Ranger (Zheng et al., 2017) for 10X Genomics, and mappa for the ICELL8 system. To date, many methods for quantifying gene expression from RNA-seq library have been proposed; however, a consensus has not yet been reached on the best quantification method for all data (Costa-Silva, Domingues & Lopes, 2017; Vieth et al., 2019; Wu et al., 2018). Moreover, since each quantification method has different measurable genes, conventional analysis methods for cell clustering using gene expression quantified via only one method are strongly biased.

After quantification, feature selection or dimension reduction is required to analyze the gene expression profile because it has a large number of rows and columns (cells and genes). Additionally, single-cell analysis involves some differences among cells (e.g., different gene expression values derived from different phases in the cell cycle and errors in measurement, such as missing values), even if these cells belong to the same cell type. For these reasons, we need a robust and data-driven clustering method for features that reflect variation in gene expression within individual cells. To date, many unsupervised feature selection, dimensionality reduction, and clustering methods have been developed (Sun et al., 2019; Kiselev, Andrews & Hemberg, 2019; Freytag et al., 2018). In particular, SC3 (Kiselev et al., 2017) and Seurat (Satija et al., 2015) are useful data-driven analysis tools for single cells. SC3 is a clustering method that uses an ensemble of multiple analysis results using the algorithm based on k-means. Seurat is a method to analyze single cells, and Louvain clustering is used in cell clustering. Matrix factorization, as a method of unsupervised learning, is another efficient method for cell clustering and is excellent in data dimension reduction or the extraction of latent factors. In particular, non-negative matrix factorization(NMF) (Lee & Seung, 1999) is a suitable method for dimension reduction to extract the features of gene expression profiles because NMF interprets the data as a superposition of the gene functions and cell characteristics. NMF factorizes a matrix into multiple matrices (basis and coefficients) under the constraint that all elements are non-negative. The product of these matrices includes an approximate matrix of the input matrix. NMF is applied to various real data because of its non-negative feature and has been adapted to microarray data for clustering or feature extraction (Brunet et al., 2004; Zheng et al., 2011; Nik-Zainal et al., 2012; Zhang et al., 2012) and scRNA-seq data (Zhu et al., 2017; Wu et al., 2020). Especially, Shao & Höfer (2017) used LSNMF, using the projected gradient method to optimize the objective function (Lin, 2007), for single-cell clustering. By performing NMF, we can analyze more details of genes and cells (e.g., finding marker genes and performing unsupervised cell clustering) with the feature matrices extracted.

However, these clustering methods do not consider the differences in quantification methods that are upstream of this series of procedures. We usually observe that different quantification pipelines produce similar but also different gene expression profiles even when we apply them to the same RNA-seq library. Consequently, different clusters are obtained and misunderstanding might be produced. Our objective is to develop a method that integrates the gene expression profiles from different quantification methods to extract reliable feature matrices for clustering. Joint-NMF, which performs multiple NMF with a shared matrix, is one of the most suitable methods for integrating such different but potentially similar data. Joint-NMF has also been studied for genomic applications, Wang, Zheng & Zhao (2015) simultaneously decompose multiple transcriptomics data matrices. Zhang et al. (2012), Yang & Michailidis (2016), Duren et al. (2018), Jin, Zhang & Nie (2020) integrated heterogeneous omics multi-modal data (e.g., DNA methylation, miRNA expression, and gene expression) to detect modules, and Fujita et al. (2018) also integrated multi-omics data to discover biomarkers with a combination of Joint-NMF and pathway analysis. These methods decompose the data into a shared matrix of “genes × modules” and some dedicated matrices of “modules × samples” for each omics data. Genes are regarded as common variables across data matrices and samples are different. In contrast, we decompose the data into some matrices of “genes × modules” and a shared matrix of “modules × cells” because the cells included in the gene expression profiles are perfectly the same in our problem setting.

In this article, we propose SC-JNMF, a novel unsupervised clustering method using NMF to eliminate the differences in quantification methods and extract the common factors over multiple gene expression profiles. The features of the data can be decomposed into gene-derived factors that contain bias dependent on each quantification method and common cell-derived factors, and cell clustering can be performed based on these common factors to obtain more essential biological information. To our knowledge, this study is the first to incorporate multiple quantification methods into the clustering analysis of scRNA-seq data.

Materials & Methods

The outline of SC-JNMF

We proposed SC-JNMF, a method that extracts latent factors from different gene expression profiles at the same time using joint-NMF, which can express matrix data as the product of lower-dimensional matrices, one of which is shared. SC-JNMF extracted the latent factors in different gene expression profiles using a similar approach to NMF and used them for cell clustering and gene analysis.

The outline of our method is showed in Fig. 1. We created two different gene expression profiles using different quantification methods from the same RNA-seq library. Then, we extracted the common factor using joint-NMF and extended it to perform multiple NMFs in parallel with two different basis factors W1, W2 (derived from the different methods) and shared factors H (derived from the original RNA-seq library). Finally, we performed cell clustering using these extracted factors and any appropriate clustering methods, such as hierarchical clustering. Although the conventional NMF clustering methods that perform clustering directly with the factorized matrix strongly depend on the rank, our method performs robust clustering by using hierarchical clustering.

Figure 1 A workflow of SC-JNMF.

(i) Creating multiple gene expression profiles using different quantification methods. (ii) Extracting the common factor among these gene expression profiles from the same RNA-seq library. (iii) Cell clustering using hierarchical clustering with the extracted common factor.

Quantification and normalization

As the first step of SC-JNMF, we quantified gene expressions with different quantification pipelines, and obtained a set of gene expression profiles. Parameters and references could be set as recommended by each quantification pipeline.

For preprocessing before performing joint matrix factorization, we applied a gene filter to the input data, similar to the SC3 method (Kiselev et al., 2017), and then log2(x + 1) transformed the data. Finally, we normalized the data so that the sum of each gene L1 norm was 1 because it should be compared the relative expression values between cells for each gene to perform the cell clustering. This L1 normalization is possible to consider smaller expression values than using the L2 norm and to keep the values non-negative.

Joint-NMF

We considered the given data as a non-negative matrix D∈R+N×M. Non-negative matrix factorization found the basis matrix W∈R+N×k and coefficient matrix H∈R+k×M, where all the elements were non-negative, such that these matrix products approximated the input data matrix. (1) D≈WH

NMF minimized the distance between matrix D and matrix WH. Here, we considered a Euclidean norm distance. Thus, the objective function that NMF minimizes was as follows: (2) minW,H≥0L:=∥D−WH∥F2

Here, we considered a simultaneous matrix factorization of two matrices using NMF. Given the two input data as non-negative matrices D1∈R+N1×M and D2∈R+N2×M, joint-NMF found the basis matrices W1∈R+N1×k,W2∈R+N2×k corresponding to each approximated input matrix and the common coefficient matrix H∈R+k×M, minimizing the distances between the given matrices and the approximated matrices. In our proposed method, we added the L1 norm constraint to this objective function for H so that the factorized shared matrix was sparse. Thus, SC-JNMF found the matrix W1, W2, H that minimized the following objective function: (3) minW1,W2,H,λn≥0L:=∥D1−W1H∥F2+λ1∥D2−W2H∥F2+λ2 ∑k∥Hk,∗∥1

where, λ1 is a balance parameter for the losses of reconstruction matrices and λ2 is a parameter for row vector sparsity regularization of shared factorized matrix.

We applied a multiplicative update algorithm to optimize the objective function same as conventional NMF. By applying Jensen’s inequalities to the first and second terms of the objective function, the function to be minimized could be rewritten as follows: (4) minW1,W2,H,λn≥0L:= ∑i,j|D1i,j|2−2D1i,j ∑kW1i,kHk,j+∑kW1i,k2Hk,j2c1i,j,k+λ1 ∑i,j|D2i,j|2−2D2i,j ∑kW2i,kHk,j+∑kW2i,k2Hk,j2c2i,j,k+λ2∥H∥1

where, (5) c1i,j,k=W1i,kHk,j ∑k′W1i,k′Hk′,j

(6) c2i,j,k=W2i,kHk,j ∑k′W2i,k′Hk′,j

We found each element of W1, W2, and H that minimized the objective function by performing partial differentiation. (7) ∂L∂W1i,k=∑j−2D1i,jHk,j+2W1i,kHk,j2c1i,j,k

(8) ∂L∂W2i,k=λ1 ∑j−2D2i,jHk,j+2W2i,kHk,j2c2i,j,k

(9) ∂L∂Hk,j=∑i−2D1i,jW1i,k+2W1i,k2Hk,jc1i,j,k+λ1 ∑i′−2D2i′,jW2i′,k+2W2i′,k2Hk,jc2i′,j,k+λ2

The objective function are minimized when these are 0. Thus, the variable updates became: (10) W1=W1D1H⊤HW1H⊤⊤

(11) W2=W2λ1D2H⊤λ1HW2H⊤⊤

(12) H=HD1⊤W1⊤+λ1D2⊤W2⊤−λ2/2W1⊤W1H+λ1W2⊤W2H

Applications

Cell clustering

The factorized matrices were used to perform highly accurate clustering. In our proposed method, we used hierarchical clustering (Ward’s method (Ward, 1963), implemented in SciPy (Virtanen et al., 2020)). In this study, the adjusted Rand index (ARI, implemented in scikit-learn (Pedregosa et al., 2011)) was used to evaluate clustering performance.

Given a set of n elements (i.e., cells) S = {o1, …, on}, and supposing that U = {u1, …, uR} and V = {v1, …, vC} represent two different partitions of S, define the following:

1. a, the number of cell pairs which are assigned to the same class in both U and V

2. b, the number of cell pairs which are assigned to different classes by both U and V

The Rand index was calculated by (13) RI=a+bn2

In addition, the ARI considered adjusting measures of clustering accuracy for chance. The ARI was defined using the following formula: (14) ARI=∑ijnij2− ∑ini⋅2 ∑jn⋅j2/n2n2− ∑ini⋅2 ∑jn⋅j2/n2

where, nij = |Ui∩Vj|, ni⋅ = ∑jnij and n⋅j = ∑inij.

Gene analysis

Our proposed method was used not only for cell clustering but also for gene analysis using the extracted factors. The factors in the coefficient matrix that showed higher values in specific clusters than those in other clusters indicated latent factors of the cluster. Therefore, the same factors in the basis matrices also showed latent factors of the cluster, and the genes that showed higher values in the basis matrices reflected characteristic features of the cluster, in other words, marker genes.

Results

To assess the accuracy of our method, we performed cell clustering using five scRNA-seq datasets and one bulk RNA-seq dataset. In this clustering, we estimated the optimal parameters (the ranks in matrix factorization) using the trade-off relationship between sparseness and loss. We also evaluated the effect of the combinations of quantification methods and the sample size for the clustering accuracy. Additionally, we analyzed more details of the factorized matrices by showing their relevance to marker genes.

Dataset

We used six different datasets including RNA-seq library and quantified gene expression levels measured using each method in previous studies (Table 1). Only Monaco dataset is a bulk RNA-seq, the others are scRNA-seq libraries. Because Monaco dataset is a set of sorted cell types, it is suitable for the evaluation of clustering accuracy. CellBench is a set of datasets designed to benchmark various single-cell data analysis. In this study, we used the dataset named “sc_10x_5cl”, single cells from the mixture of five cell lines. In advance, we removed any cell types that were unclear in previous studies.

Table 1 Dataset and quantification methods in previous studies.

Dataset (citaion)	Tissue/process	Quantification	Reference	The number of cells	The number of classes	
Treutlein (Treutlein et al., 2014)	Lung epithelium	TopHat v2.0.8, Cufflinks v2.0.2	mm10	80	5	
Pollen (Pollen et al., 2014)	Brain	TopHat v2.0.4, RSEM v1.2.4	hg19	259	10	
Segerstolpe (Segerstolpe et al., 2016)	Pancrease	STAR v2.3.0e, rpkmforgenes	hg19	2166	12	
Xin (Xin et al., 2016)	Pancrease	CLC Bio Genomics Workbench v7.0	GRCh37	1492	4	
Monaco (Monaco et al., 2019)	Immune cells	Kallisto v0.43.1, Tximport v1.6.0	GENCODE Human 26	114	10	
CellBench (sc_10x_5cl) (Tian et al., 2019)	Lung adenocarcinoma cell lines	scPipe v.1.3.0	GRCh38	3918	5	

To perform our method SC-JNMF and compare the accuracy of the clustering methods, We alternatively quantified gene expression values in each cell from the RNA-seq library using Salmon(v1.0.0) with GENCODE references and annotations (Mouse Release 21/ Human Release 32). Only CellBench dataset was quantified by Using Salmon with Alevin (Srivastava et al., 2019), kallisto with BUStools and STAR with Cell Ranger to process barcodes in short reads.

In addition, for the first 4 datasets, we also prepared other gene expression profiles using kallisto(v0.46.2), STAR(v2.7.3), to evaluate our method for the combinations of quantification methods and the sample size.

Settings of clustering methods

We compared the accuracy of cell clustering using our proposed method to that obtained using other major unsupervised clustering methods, including LSNMF (Lin, 2007; Zitnik & Zupan, 2012), SC3 (Kiselev et al., 2017), and Seurat (Satija et al., 2015).

For our proposed method, each run consisted of ten runs and extracted an ARI score of the top combined rank in terms of sparseness and loss (the top ranking of sparseness is maximum and the top ranking of loss is minimum). The rank was determined according to the results of our experiment described in the Supplemental Information and Fig. S1 as follows: Treutlein, k = 5; Pollen, k = 8; Segerstolpe, k = 25; Xin, k = 16; Monaco, k = 8; CellBench, k = 5. In addition, we determined the regularization parameter λ1 = |geneset1|/|geneset2|. Hierarchical clustering could not determine the number of clusters; therefore, we set the same number reported in previous studies in advance.

In the LSNMF method, similar to our joint-NMF, classification by hierarchical clustering was performed using a matrix of factors of the cells, and the rank and the number of clusters were set to the same values as ours. SC3 method was performed with the default parameters. For LSNMF and SC3, each run was performed ten times, considering random initial values. For Seurat, we plotted only the highest ARI score in the runs as the resolution parameter in the FindClusters function was increased from 0 to 1 in 0.1 steps. The parameters of the FindNeighbors function were set to the default values (dims = 1:10, k.param = 20).

Accuracy of cell clustering

Figure 2 shows the ARI score of the original (quantified previously) and alternative (quantified using Salmon) data for each clustering method. We ran a two-way experiment using λ2 = 1 and λ2 = 10.

Figure 2 The accuracy (ARI) of cell clustering in each method.

For comparison methods, we performed experiments using the original gene expression profile (quantified previously) and alternative gene expression profile (quantified using Salmon). For our proposed method, we performed the experiment in two ways with different regularization parameters (λ2 = 1 and λ2 = 10).

In the Pollen, Xin, and CellBench datasets, our proposed method performed accurate cell clustering. In contrast, in the Treutlein, Segerstolpe, and Monaco datasets, the ARI score of the proposed method was higher than that of LSNMF, but it was not the highest score. Additionally, the resulting clusters of our method were stable compared to the conventional NMF method (except for the CellBench dataset).

Comparison of quantification methods in SC-JNMF

SC-JNMF performed cell clustering using one or two gene expression profiles. In this study, we compared the combinations of quantification methods (kallisto, STAR, Salmon) by performing cell clustering and calculating ARI using SC-JNMF. To evaluate the impact of “joint” NMF, we also performed factorization and clustering using a single gene expression profile with setting to the parameter λ1 = 0. We ran SC-JNMF repeatedly for ten trials with random initial values and the parameter λ2 = 10.

Figure 3 shows the ARI of the cell clustering using SC-JNMF in each quantification method and their combination. As a result, for each dataset, the accuracies of kallisto & STAR and Salmon & STAR combination outperformed using the single quantification method. The accuracy of Salmon & kallisto combination was lower than that of other combinations.

Figure 3 The accuracy (ARI) of cell clustering in each quantification method and their combination.

Additionally, we counted the frequency of correlation coefficients for each cell and each gene in each dataset and combination (Fig. 4). In Salmon & kallisto combination, the correlation coefficients were inclined toward a higher value than the other combinations. These results indicated that SC-JNMF had a lower accuracy when using similar gene expression profiles (e.g., Salmon & kallisto combination). Therefore, two gene expression profiles with different properties, having a lower correlation of cells and genes, were suitable for SC-JNMF.

Figure 4 Histograms ( log10 scale) of correlation coefficients in each gene and cell.

Size effect for SC-JNMF and NMF

To further understand the benefit of NMF-based methods, the effect of sample size (the number of cells in the RNA-seq library) against clustering accuracy was evaluated. We randomly subsampled the 25%, 50%, and 75% cells from gene expression profiles, and ran SC-JNMF (λ2 = 10) and LSNMF. Each trial was performed ten times.

Figure 5 shows the ARI of the cell clustering using LSNMF and SC-JNMF for each dataset and subsample rate. As a result, for each dataset, the accuracies of both LSNMF and SC-JNMF were increased depends on the subsample rate. This result indicated that the methods based on NMF benefit more from a dataset including a larger number of cells.

Figure 5 The accuracy (ARI) of cell clustering against subsample rate.

Because the original quantification method of Segerstolpe dataset is STAR, the plots for the “star” and the combination of “original & star” are blank. For the same reason, “kallisto” and “original & kallisto” are blank for the Monaco dataset.

Gene analysis using factorized matrix

We found marker genes of the Xin dataset using factorized matrices. The Xin dataset contained data on 1492 single cells and four classes (alpha, beta, delta, and PP) in the pancreas. We showed the results in Fig. 6. The factor of the coefficient matrix showed some characteristic patterns in each cell cluster (e.g., Factor 2 and Factor 5 showed higher values in delta and PP cells) (Fig. 6A). Next, we calculated the correlation of factors between the basis matrices in the common genes (Fig. 6B). The factors in the basis matrices showed a similar tendency. We also showed the loadings of marked factors for delta cells in the coefficient matrix in Fig. 6C. Almost all genes showed similar factor loadings between base1 and base2; however, we confirmed that some of the genes only observed in either gene expression profile also had high values. We also showed the marker genes of delta cells detected using Scanpy Wolf, Angerer & Theis (2018) in the scatter plot. The factor loadings of these genes tended to be higher than those of others, regardless of whether the gene was observed in both gene expression profiles or not. All factors of Xin dataset were shown in Fig. S2.

Figure 6 Gene analysis of the Xin dataset using SC-JNMF.

(A) A heatmap of the common coefficient matrix generated using SC-JNMF. (B) The correlation of the factor loadings in genes common to the basis matrix1 and basis matrix2. (C) “factor 2” loadings of each gene in base1 and base2 and marker genes of delta cells in original and alternative gene expression profiles detected using Scanpy (Wolf, Angerer & Theis, 2018).

Discussion

Thus far, we have shown the possibility that our unsupervised clustering method had high accuracy when using the differences in gene expression quantification methods compared with previous studies. However, the proposed method showed worse accuracy than SC3 in the original Treutlein dataset, as well as Seurat in the original Segerstolpe dataset, and showed differences in the accuracy due to the differences in regularization parameters in the Pollen dataset. Treutlein and Pollen datasets have fewer cells than the others, which was one of the most important features of our experiments. Dataset size is an important factor (common to machine learning approaches) for high accuracy, and this characteristic is also applied to our method.

We showed the effective combinations of quantification methods by comparing the accuracy of cell clustering for each combination. In particular, we suggested that the combination of gene expression profiles that have similar properties (e.g., Salmon & kallisto) had lower accuracy in SC-JNMF. Compared to the combination, including STAR that maps RNA-seq reads to a reference genome, Salmon and kallisto are similar methods in the quantification algorithm, as these are alignment-free quantification methods. Therefore, these gene expression profiles have similar properties. In SC-JNMF using similar gene expression profiles, it is difficult to separate common factors derived from cells (H) and factors derived from genes (W1, W2). Meanwhile, although it has been reported that the pseudo-alignment method loses many reads, leading to a lower mean expression (Vieth et al., 2019), it is possible to improve the clustering performance by incorporating them into our method with the other quantification method.

We presented more details about the characteristics of the factorized matrix and the relationships of marker genes. The coefficient matrix showed characteristic factors in each cell cluster, and both basis matrices had similar factors. The marker genes of a cell cluster showed high factor loadings in the basis matrices that characterized a specific cell cluster in the coefficient matrix, regardless of including both gene expression profiles or only one. In other words, factor loadings in basis matrices that characterize a specific cell cluster in the coefficient matrix are related to the marker genes for that cluster. This result suggested that genes showing high factor loadings in the basis matrices probably had some important features in the cluster with high factor loadings in the coefficient matrix. In particular, we should pay particular attention to those genes observed only in either gene expression profile because they are not considered in conventional methods.

In summary, our method is effective in the following cases.

• The number of cells is sufficient large.

• Different quantification methods yield gene expression profiles with different characteristics.

Meanwhile, there are some limitations. As shown in the result of CellBench dataset, the number of cells is quite large and the nature of the cell type is well defined, jointless NMF gives sufficiently good results. In such a case, our joint-NMF may deteriorate the stability of the solution. Moreover, we also observed cases that the joint-NMF worsened the accuracy (“salmon & kallisto” combination in Pollen dataset, Fig. 3), although the effect of the “joint” was either better or unchanged in many cases. It should be avoided to input expression profiles quantified by similar approaches.

Conclusion

We proposed SC-JNMF, which performs cell clustering using common factors extracted from multiple gene expression profiles quantified using different methods. As a result, it is possible to perform robust analysis compared with the case in which only a single quantification method is used because it is unnecessary to consider the differences in gene expression profiles. The accuracy (ARI) of cell clustering obtained using our method was higher than that of other major clustering methods. Additionally, we showed that the combination of different quantification methods increases the accuracy of cell clustering compared to that of a similar quantification. Moreover, we showed the details of the extracted factors. The genes characteristic to specific cell groups (marker genes) showed remarkable factor loadings in terms of the factorized matrices; in other words, these results suggest a potential for identifying important genes in the dataset. Some genes may not be counted depending on the quantification methods used; they can be detected using multiple gene expression profiles generated using different quantification methods and SC-JNMF if they are potential markers.

Supplemental Information

Supplemental Information 1 Supplemental information

Click here for additional data file.

Supplemental Information 2 Loss and sparseness of the matrix H in each rank

We performed the experiment in two ways with different regularization parameters λ2 = 1 and λ2 = 10.

Click here for additional data file.

Supplemental Information 3 Scatterplot with density for all factors decomposed by using SC-JNMF (Xin dataset)

Points show the genes and their color indicated the density (the number of neighbors).

Click here for additional data file.

Additional Information and Declarations

Competing Interests

Author Contributions

Data Availability

The authors declare there are no competing interests.

Mikio Shiga and Shigeto Seno conceived and designed the experiments, performed the experiments, analyzed the data, prepared figures and/or tables, authored or reviewed drafts of the paper, and approved the final draft.

Makoto Onizuka and Hideo Matsuda analyzed the data, authored or reviewed drafts of the paper, and approved the final draft.

The following information was supplied regarding data availability:

SC-JNMF is implemented in Python. The source code and documentation are available at GitHub: https://github.com/agis09/sc-jnmf.

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
