# Peer review of "SC-JNMF: single-cell clustering integrating multiple quantification methods based on joint non-negative matrix factorization"

_PeerJ, doi:10.7717/peerj.12087_

## Round 0.1 · original submission · Major Revisions

Your manuscript has been reviewed by three experts in the field. As you can see from their comments below, all of them raise rather fundamental criticisms on it; one of them even recommends its rejection. Thus, please read their comments carefully and revise the manuscript accordingly. All of the reviewers seem to point out some concerns on (the derivation of) the presented equations. Also, they give some concerns on the basic design of the experiment itself. I hope that your next revision will be satisfactory to the reviewers.

Reviewer 1 ·

Basic reporting

This paper develops a new method named SC-JNMF for integration of multiple quantification methods to perform stable clustering of scRNA-seq data.

Experimental design

no comment

Validity of the findings

no comment

Additional comments

Please see my comments:
1, Authors point out that there are many gene expression quantification pipelines, like cufflink, RSEM, etc, and they provide inconsistent results. But those quantification methods are designed for bulk RNA-seq data and not for single cell data. On scRNA-seq, most research just use the read count instead of FPKM, RPKM, or TPM. It is not clear what is the benefit of using those quantification methods on single cell data.
2,NMF is widely used in single cell clustering analysis. Important citations are missing in introduction:
A. Shao, Chunxuan, and Thomas Höfer. "Robust classification of single-cell transcriptome data by nonnegative matrix factorization." Bioinformatics 33.2 (2017): 235-242.
B. Duren, Zhana, et al. "Integrative analysis of single-cell genomics data by coupled nonnegative matrix factorizations." Proceedings of the National Academy of Sciences 115.30 (2018): 7723-7728.
C. Jin, Suoqin, Lihua Zhang, and Qing Nie. "scAI: an unsupervised approach for the integrative analysis of parallel single-cell transcriptomic and epigenomic profiles." Genome biology 21.1 (2020): 1-19.
3, The update role in Eq. (7-9) looks incorrect to me. Please check the equations carefully. For example, D1 and H are cannot be directly multiped since the dimensions are not matching.

Reviewer 2 ·

Basic reporting

see General comments

Experimental design

see General comments

Validity of the findings

see General comments

Additional comments

The authors proposed a joint-NMF based method to integrate expression matrix with different quantification methods and applied join-NMF for cell clustering.

Although the idea of the shared matrix of “modules x samples” is already applied for multimodal data (https://www.nature.com/articles/s41467-020-20430-7 for example), such idea is not applied for multiple expression data with different quantification methods.

Although the integration of such expression data is interesting topic, I have several concerns for publication.

Major comments
1. In my opinion, the experimental design is not appropriate for clustering validation. I think the cell-type (label) of the Segerstolpe dataset is determined with the original expression data, for example. In such a case, it is obvious and not interesting that the ARI values with original expression data (or data with similar quantification methods) are better than those with Salmon and kallisto. Therefore, the conclusion that “ARI tended to be low only when using Salmon as the quantification method” (line.205) is inadequate. Overall, I think this bias makes us difficult to interpret the effects of different quantification methods and different clustering methods.

2. To overcome the above problem, I recommend the authors use the bulk RNA-seq dataset of sorted cell types such as https://dice-database.org/downloads#info_anchor and https://www.sciencedirect.com/science/article/pii/S2211124719300592 for clustering evaluation.

3. The scRNA-seq dataset in this article is not the latest. I recommend that the author add the evaluation with the latest scRNA-seq data because the quality and properties of the data are improving.

4. I have a concern about the merit of joint-NMF. In the current results, I cannot follow the effect of “joint” because the ARI values result from comprehensive procedures. I want to know the impact of “joint”-NMF by comparing the ARI values for such as
- Combining two expression matrices into one matrix, using general NMF and applying the same clustering method.
- Using only one expression matrix, using general NMF, and applying the same clustering method.
- Averaging the expression matrix, using general NMF, and applying the same clustering method.

Minor comments
Line.57: It is inappropriate to say “without mapping” about Salmon and kallisto because these methods roughly map reads to the transcripts with k-mer. It is appropriate to say “without alignment”.

Line.132: The authors say “log2 transformed the data”, but I wonder it produce negative values ?

Line.137: The dimension of D2 and W1 are wrong?

Line.142: The notations such as W_{1k} are confusing because the authors use H^{kj} in the following description. I want the authors to rewrite the united notation.

Line.143: I think lambda_1 is not related to “column vector sparsity regularization”.

Line.180: The reference to the table does not work well.

Line. 188: I cannot follow the detailed clustering setting. For example, I think the authors used FindNeighbors() and FindClusters() of Seurat Clustering, and I want to know the setting and effects of “dims” parameter. In addition, the other clustering method is described in the result section that makes me difficult to follow the result. Please divide the method and result into different sections.

Reviewer 3 ·

Basic reporting

The language needs to be improved. page 5 line 159, what is "the ratio of the total of".

Also some terminologies are not standard. The authors used "biased" in many places, where "variation" is more appropriate. For example, in page 2 line 70, "we need ... that reflect individual cell biases", but the author meant the gene expression variation among individual cells. To refer to a RNASeq dataset, "RNA-seq library" is more commonly used than "RNA-seq reads". In the factor analysis literature, "latent factor" is more common than "latent feature" used by the authors. For example, on page 3 line 108-110, "features" was used in many places in this sentence but they have different meanings. The authors could use "cell-derived factors" instead of "features". "Rank" is more common than "Rank number". On page 6 line 228, the authors could use "factor loadings" instead of "factor values".

Experimental design

On page 4 line 133, the author normalized gene expression by L1 norm instead of L2 norm. The author could explain the rationale of that.

Equation(3), the author should clarify which matrix norm is used for W and H. Also in their experiments, \lambda_2 and \lambda_3 are always set to be zero. So they don't actually need to include these two terms. If they include, they need to show how to select these two hyperparameters.

It's unclear how to derive equations 7-9 from 4-6.

As quantification by STAR has better performance than others, \lambda_1 should have important role on the performance of joint NMF. The authors fixed \lambda_1 to match the number of genes, but it would be interesting to show how the performance varies with \lambda_1.

The author mentioned the sample size can affect the clustering performance (page 7 line 240), but it's unclear from the manuscript in which case their method is more beneficial. The authors could subsample the data set and compare their joint NMF with single NMF varying data size.

The method of choosing the rank is interesting but lacks justification. why the rank when the sparseness hits the elbow point is a good choice.

Validity of the findings

The improvement of joint NMF on two quantification methods compared with only using STAR is not clearly shown. Fig 3 only has aRIs for combination of two methods, it'd better if including the result only using one of the quantification methods and including all three methods.

The author stated that using similar gene expression profiles is worse than combining different gene expression profiles. However, different gene expression profiles have different performance on their own. The reason combining with STAR has better performance is likely because STAR is a better quantification method. Instead of comparing different combinations directly, it makes more sense to compare the performance of each combination with the performance of each individual method. Also line 248 on page 7 is quite confusing.

In figure 5C, several genes have quite different weights in two quantification methods. It would be interesting to provide several examples with further diagnosis.

---

## Round 0.2 · Major Revisions

Your revised manuscript has been reviewed by two of the three original reviewers (the remaining one has not responded to our invitations). As you can see from their comments below, one of them is now satisfied with the revision while the other requests further revision. Since I basically support this reviewer's opinion, please re-revise the manuscript as requested. Thanks for your patience, in advance.

Reviewer 1 ·

Basic reporting

no comment

Experimental design

no comment

Validity of the findings

no comment

Additional comments

Authors have solved all my comments, I have no further comments.

Reviewer 2 ·

Basic reporting

see comments

Experimental design

see comments

Validity of the findings

see comments

Additional comments

The authors tried to resolve all of my concerns, but I still have a few concerns.

1. I cannot follow why the authors used Monaco and CellBech dataset only in Fig.2. In my opinion, these datasets will be useful for other analyses (such as Fig.3).
2. The ARI values of “salmon and kallisto” are lower than those of “kallisto” or “salmon” for the Pollen dataset. In my opinion, it is important to analyze and discuss this case for clarifying the usefulness of “joint” NMF.
3. It is difficult to see and compare the values of different methods in Fig.5. I want the authors to revise Fig.5 so that the boxplot are arranged for the same dataset and same subsample rate, likewise Fig.3.

---

## Round 0.3 · accepted · Accept

Your re-revised manuscript has been reviewed by the original reviewer who gave several comments on its previous version. As you can see from his/her comments below, the reviewer is satisfied with your revision, understanding that one of the points was hard to answer clearly. Thus, I am happy to inform you that I will recommend its acceptance to the Editor-in-Chief.

Reviewer 2 ·

Basic reporting

no comment

Experimental design

no comment

Validity of the findings

no comment

Additional comments

The authors tried to resolve all of my concerns, and now, I recommend that it be accepted for publication.

I understand it isn't easy to clarify the reason for comment.2, and I hope the authors will find the reason in future research.

---

## Author Rebuttal · Round 0.3

August 3rd, 2021

Dear Editors

  We thank the editor and all the reviewers very much again, for their constructive comments on our revised manuscript, "SC-JNMF: Single-cell clustering integrating multiple quantification methods based on joint non-negative matrix factorization" by Shiga et al. In response to the comments, we performed additional experiments and revised the manuscript. Please check the point-by-point response to the reviewers' comments in the following pages.

  We deeply appreciate for the opportunity to improve our manuscript according to the reviewers' valuable comments and suggestions. We hope that the revised version is suitable for publication.

Shigeto Seno, Ph.D.

Associate Professor of Graduate School of Information Science and Technology, Osaka University

On behalf of all authors.

# Reviewer 1

## Basic reporting

no comment

## Experimental design

no comment

## Validity of the findings

no comment

## Additional Comments

Authors have solved all my comments, I have no further comments.

# Reviewer 2

## Basic reporting

see comments

## Experimental design

see comments

## Validity of the findings

see comments

## Additional Comments

The authors tried to resolve all of my concerns, but I still have a few concerns.

1. I cannot follow why the authors used Monaco and CellBench dataset only in Fig.2. In my opinion, these datasets will be useful for other analyses (such as Fig.3).

[Our response:]:

We thank the reviewer for the suggestion. We added the results of Monaco and CellBench datasets in Fig.3, Fig.4 and Fig.5, and revised the manuscript.

There were some difficulties for us to execute the quantification pipelines including

"demultiplexing barcodes". In CellBench dataset, RNA-seq reads of all cells are stored in a single file using barcoding technology. In the dataset we used at the first submission, RNA-seq reads were stored in separate fastq file for each cell.

Now we can handle the quantification pipelines including demultiplexing.

**2. The ARI values of "salmon and kallisto" are lower than those of "kallisto" or "salmon" for the Pollen dataset. In my opinion, it is important to analyze and discuss this case for clarifying the usefulness of "joint" NMF.**

[Our response:]:

We thank the reviewer for this very important comment. For the past month, we have been examining various situations to answer this comment. However, we could not come up with a clear answer.

The reason for this may be the effect of the loss function of NMF, the update algorithm, and the initial values, in addition to the dataset dependence. For "salmon & kallisto" in Pollen's dataset, our Joint-NMF can give a high ARI, however, the variance of the ARI is large. In this case, it may be difficult to obtain stable solutions by joint-NMF. This can be improved by adjusting some parameters, but originally, we do not assume to input two similar expression profiles. In an extreme case, there would be no joint effect if the same expression profiles are used as inputs.

Based on the reviewers' comments and additional experiments, the cases in which our method is useful can be summarized as follows:

1. The number of cells is sufficient large (the accuracy decreases when the subsample rate decreases).

2. Different quantification methods yield expression profiles with different characteristics.

These discussions had been added to "Discussion" as limitations.

NMF is a widely used method, but the regularization techniques and update algorithms are still being studied. This comment is a very essential and difficult issue, so we would like to work for it as a future work.

**3. It is difficult to see and compare the values of different methods in Fig.5. I want the authors to revise Fig.5 so that the boxplot are arranged for the same dataset and same subsample rate, likewise Fig.3.**

[Our response:]:

The reason for the arrangement of the previous Fig.5 was that we wanted to show that the higher the subsample rate, the better the accuracy for both LSNMF and our Joint-NMF method.

Now we followed the comments and revised Fig.5.